# Initial Rotor Position Detection for Permanent Magnet Synchronous Motor Based on High-Frequency Voltage Injection without Filter

**Zhiqiang Wang [1], Bo Yao [2], Liyan Guo [2], Xuefeng Jin [1], Xinmin Li [2] and Huimin Wang [2,***

[1] School of Artificial Intelligence, Tiangong University, Tianjin 300387, China;
wangzhiqiang@tiangong.edu.cn (Z.W.); jinxuefeng@tiangong.edu.cn (X.J.)

[2] School of Electrical Engineering and Automation, Tiangong University, Tianjin 300387, China;
1831045303@tiangong.edu.cn (B.Y.); guoliyan@tju.edu.cn (L.G.); lixinmin@tju.edu.cn (X.L.)

* Correspondence: wanghuimin@tju.edu.cn; Tel.: +86-1350-207-6811

**Abstract:** The accurate initial rotor position of a permanent magnet synchronous motor (PMSM) is necessary for starting the motor, and for the position sensorless control method adopted by a PMSM control system under some working conditions. This paper presents a new method to detect the initial rotor position of a permanent magnet synchronous motor (PMSM). The method does not need a low-pass filter, and has strong robustness and a simple calculation method. According to the relationship between high-frequency current response and rotor position angle $\theta$, the rotor position angle can be obtained by arctangent and linear formulae. Finally, the magnetic polarity of the rotor is distinguished according to the change of inductance. In this method, the arctangent function is used to eliminate the filtering process and reduce the influence of the parameter deviation of the motor system on the detection accuracy of the initial position. The experimental results verify the correctness of the theoretical analysis and the effectiveness of the method.

**Keywords:** permanent magnet synchronous motor; rotor position detection; high frequency voltage injection; magnetic saturation

## 1. Introduction

The permanent magnet synchronous motor has the advantages of simple structure, low loss, high power factor and high efficiency, so it has been widely used in the fields of industrial robots, aerospace and other fields. In order to achieve high-performance control, it is necessary to accurately detect the initial rotor position [1–9]. The running performance of the motor will be decreased and a smooth start-up process will not be achieved unless the initial rotor position of the permanent magnet synchronous motor is accurate. Therefore, the initial position of the rotor needs to be accurately estimated in order to realize the high-performance control [1–9].

There are two kinds of calculation principle in the position sensorless control of a permanent magnet synchronous motor (PMSM). The first one is based on the fundamental mathematical model of the PMSM and the second is based on the salient characteristics of the motor. Among the above-mentioned methods, the first method usually extracts the position signal from the back electromotive force (EMF), according to the relationship between back EMF and rotor position. The EMF can be obtained directly from the voltage equation. This method is very simple, but it needs accurate motor parameters [10]. However, it is very difficult to calculate the back EMF based on the voltage equation because the back EMF of the motor will be small when the speed drops to zero. Therefore, it is necessary to make use of the salient polarity of the motor. In order to obtain significant salient polarity, the signal injection

method is usually used to obtain the high-frequency response current, which contains the rotor position information. The high-frequency signal injection method is based on the salient pole effect of the motor, which can effectively estimate the initial rotor position of the permanent magnet synchronous motor. According to the nature of the injection signal, it can be divided into a rotating high-frequency signal injection method [11] and a high-frequency pulse signal injection method [12]; according to the position of the injection signal of the whole motor control system, it can be divided into a high-frequency voltage signal injection method [12–14] and a high-frequency current signal injection method [15]. Among these, the representative methods include a voltage pulse vector injection method [16,17] and a high-frequency signal injection method [17,18]. In reference [19], a series of voltage pulses with equal amplitude and different directions are applied to the motor, and the initial position of the rotor is estimated by detecting and comparing the corresponding stator current, which is the basic idea of the voltage pulse injection method. The theoretical error of this method is small, but it is difficult to determine the pulse width and amplitude and to implement this approach, and the error also affects the accuracy of position estimation. In reference [20], initial rotor position is estimated from a spatial saliency tracking estimator. Magnet polarity is identified simultaneously by using the fundamental harmonic of rotor position in the second harmonic frame of the carrier frequency due to d and q-axis saturation saliencies for both rotating and pulsating voltage carrier injection. However, this method uses a low-pass filter, and the algorithm is complex and difficult to implement. In reference [21], the high-frequency square wave signal injection method is adopted. This method does not need to use a filter to extract the high-frequency response current, which increases the system bandwidth. However, the high-frequency square wave signal has the disadvantages of strong sensitivity to the measurement error, and is easily affected by the sampling delay and the nonlinear effect of the inverter. In reference [22], according to the characteristics of the phase induction of Inner Permanent Magnet Synchronous Motors (IPMSMs), the corresponding relationship between the rotor position and the phase inductance is obtained. In this method, the rotor position information is obtained easily, with several basic math operations and logical comparisons of phase inductances, without any coordinate transformation or trigonometric function calculation. However, the algorithm is sensitive to disturbance and has poor robustness.

First of all, this paper studies the conventional high-frequency sinusoidal voltage injection method [23,24]. The principle of this method is to inject a high-frequency sinusoidal voltage signal into the α-β-axis, and then generate the high-frequency response current. Finally, the initial position of the rotor is obtained by using the relationship between the amplitude of the high-frequency response current and the position of the rotor. However, this method requires bandpass filters when demodulating the amplitude of the high-frequency response current, and the use of the filters will delay the rotor position and cause errors. When calculating the rotor position, the arcsine formula is used. This method includes physical quantities such as inductance and injection voltage. These physical quantities will affect the calculation results and make them inaccurate.

In this paper, aiming at the shortcomings of the traditional sinusoidal signal injection method to detect the initial position of the rotor, a new method based on a high-frequency sinusoidal voltage signal injection method is proposed. The advantages of the proposed method are as follows.

(1)  There is no need to use a filter to extract the high-frequency response current.
(2)  It is less dependent on parameters.
(3)  The method is simple and effective, and occupies fewer resources of the digital control system.

Theoretical analysis and experimental results show that the method can accurately detect the rotor's initial position, and verify the feasibility and effectiveness of the method proposed in this paper.

This paper discusses the effectiveness of this method based on the following aspects. First, the mathematical model of the method is deduced and the principles of the method are described. Second, the implementation steps of the method are introduced and error analysis is carried out. Next,

the rotor polarity is judged. Finally, the effectiveness of the method is verified and conclusions are drawn through experiment.

## 2. The Principles of Rotor Position Detection

### 2.1. Mathematical Model of Permanent Magnet Synchronous Motor

Due to the fact that the mathematical model of a permanent magnet synchronous motor (PMSM) has the characteristics of multi-variability, strong coupling and nonlinearity, it is difficult to establish an accurate mathematical model for a permanent magnet synchronous motor. In order to simplify the mathematical model, the following assumptions are made: (1) neglecting the saturation of the magnetic circuit of the motor, the magnetic circuit is considered linear; (2) the hysteresis and eddy current losses of the motor are neglected; (3) the magnetic field of the permanent magnet of the rotor distributes sinusoidally in the air gap, and the back electromotive force of the motor is a symmetrical three-phase sinusoidal wave.

The PMSM schematic diagram is shown in Figure 1. Based on the above assumptions, the established model of the motor in synchronous rotation in a d-q axis system is as follows.

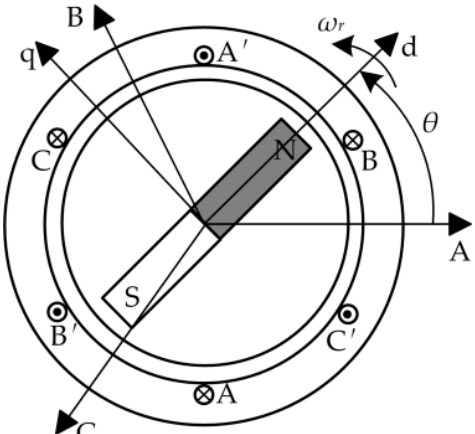

**Figure 1.** Schematic diagram of permanent magnet synchronous motor (PMSM).

The flux equation can be expressed as

$$\begin{bmatrix} \psi_d \\ \psi_q \end{bmatrix} = \begin{bmatrix} L_d & 0 \\ 0 & L_q \end{bmatrix} \begin{bmatrix} i_d \\ i_q \end{bmatrix} + \begin{bmatrix} \psi_f \\ 0 \end{bmatrix} \tag{1}$$

where $\psi_d$ and $\psi_q$ are the flux in the d and q axes, respectively; $L_d$ and $L_q$ are inductance in the d and q axes, respectively; $\psi_f$ is the flux of the permanent magnet.

The voltage equation can be derived as

$$\begin{bmatrix} u_d \\ u_q \end{bmatrix} = R_s \begin{bmatrix} i_d \\ i_q \end{bmatrix} + w_r \begin{bmatrix} -\psi_q \\ \psi_d \end{bmatrix} + \frac{d}{dt} \begin{bmatrix} \psi_d \\ \psi_q \end{bmatrix} \tag{2}$$

where $u_d$ and $u_q$ are the voltage in the d and q axes, respectively; $R_s$ is stator resistance; $i_d$ and $i_q$ are currents in the d and q axes, respectively.

The torque equation of the PMSM can be expressed as

$$T_e = \frac{3}{2}p\left[\psi_f i_q + \left(L_d - L_q\right)i_d i_q\right] \tag{3}$$

where $T_e$ is the electromagnetic torque of the permanent magnet synchronous motor and $p$ is the number of motor pole pairs. For the interior permanent magnet synchronous motor, there is $L_d \neq L_q$.

### 2.2. Principle of Initial Position Detection

In the A-B-C coordinate system of the PMSM, the voltage equation of the three-phase winding can be expressed as

$$\begin{bmatrix} u_A \\ u_B \\ u_C \end{bmatrix} = R_s \begin{bmatrix} i_A \\ i_B \\ i_C \end{bmatrix} + \frac{d}{dt} \begin{bmatrix} \psi_A \\ \psi_B \\ \psi_C \end{bmatrix} \tag{4}$$

where $u_A$, $u_B$ and $u_C$ are, respectively, the phase voltage of the A-B-C three-phase winding; $i_A$, $i_B$ and $i_C$ are the phase current of each phase winding; $\psi_A$, $\psi_B$ and $\psi_C$ are the flux linkage of each phase winding. There is

$$\begin{bmatrix} \psi_A \\ \psi_B \\ \psi_C \end{bmatrix} = \begin{bmatrix} L_A & L_{AB} & L_{AC} \\ L_{BA} & L_B & L_{BC} \\ L_{CA} & L_{CB} & L_C \end{bmatrix} \begin{bmatrix} i_A \\ i_B \\ i_C \end{bmatrix} + \begin{bmatrix} \psi_{fA} \\ \psi_{fB} \\ \psi_{fC} \end{bmatrix} \tag{5}$$

where $L_A$, $L_B$ and $L_C$ are, respectively, the self-inductance of the A-B-C three-phase winding; $L_{xx}$ is the mutual inductance of each phase winding; $\psi_{fA}$, $\psi_{fB}$ and $\psi_{fC}$ are the flux of the A-B-C three-phase winding due to the permanent magnet. There is

$$\begin{bmatrix} \psi_{fA} \\ \psi_{fB} \\ \psi_{fC} \end{bmatrix} = \psi_f \begin{bmatrix} \cos\theta \\ \cos(\theta - 2\pi/3) \\ \cos(\theta + 2\pi/3) \end{bmatrix} \tag{6}$$

where $\theta$ is the angle between the rotor permanent magnet and the A-axis, which is also the angle between the d-axis and the $\alpha$-axis, as shown in Figure 2.

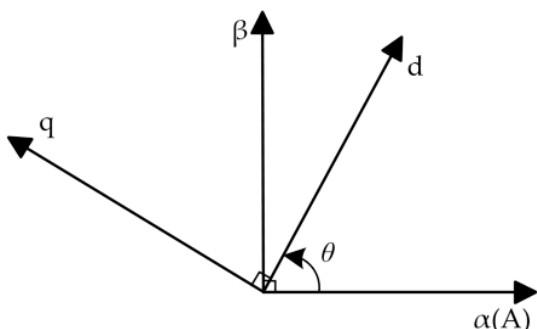

**Figure 2.** Relationship between the reference frames.

The self-inductance and mutual inductance of the stator winding are, respectively, expressed as

$$\begin{bmatrix} L_A \\ L_B \\ L_C \end{bmatrix} = \begin{bmatrix} L_0 \\ L_0 \\ L_0 \end{bmatrix} + L_2 \begin{bmatrix} \cos 2\theta \\ \cos 2(\theta - 2\pi/3) \\ \cos 2(\theta + 2\pi/3) \end{bmatrix} \tag{7}$$

$$\begin{bmatrix} L_{AB} \\ L_{BC} \\ L_{CA} \end{bmatrix} = \begin{bmatrix} L_{BA} \\ L_{CB} \\ L_{AC} \end{bmatrix} = -\frac{1}{2} \begin{bmatrix} L_0 \\ L_0 \\ L_0 \end{bmatrix} + L_2 \begin{bmatrix} \cos 2\theta \\ \cos 2(\theta - \pi/2) \\ \cos 2(\theta + \pi/3) \end{bmatrix} \tag{8}$$

where $L_0 = (L_d + L_q)/2$, $L_2 = (L_q - L_d)/2$. Due to the saliency of the interior permanent magnet synchronous motor, there is $L_d < L_q$.

Taking Equations (6), (7) and (8) into (4) and (5), then through 3/2 transformation, in the $\alpha$-$\beta$-axis, the stator voltage equation can be gained as

$$\begin{bmatrix} u_\alpha \\ u_\beta \end{bmatrix} = R_s \begin{bmatrix} i_\alpha \\ i_\beta \end{bmatrix} + \frac{d}{dt} \begin{bmatrix} \psi_\alpha \\ \psi_\beta \end{bmatrix} + w_r \begin{bmatrix} -\psi_\beta \\ \psi_\alpha \end{bmatrix} \tag{9}$$

where $u_\alpha$ and $u_\beta$ are the voltage of the the $\alpha$ and $\beta$ axes, respectively, and $\psi_\alpha$ and $\psi_\beta$ are the flux of the $\alpha$ and $\beta$ axes, respectively. When the motor is stationary, the angular velocity is zero ($\omega_r = 0$), and the voltage equation can be simplified as

$$\begin{bmatrix} u_\alpha \\ u_\beta \end{bmatrix} = R_s \begin{bmatrix} i_\alpha \\ i_\beta \end{bmatrix} + \frac{d}{dt} \begin{bmatrix} \psi_\alpha \\ \psi_\beta \end{bmatrix} \tag{10}$$

The equation of flux linkage can be expressed as

$$\begin{bmatrix} \psi_\alpha \\ \psi_\beta \end{bmatrix} = i_\alpha \begin{bmatrix} L_0 + L_2\cos 2\theta \\ L_2\sin 2\theta \end{bmatrix} + i_\beta \begin{bmatrix} L_2\sin 2\theta \\ L_0 - L_2\cos 2\theta \end{bmatrix} + \psi_f \begin{bmatrix} \cos\theta \\ \sin\theta \end{bmatrix} \tag{11}$$

In Equation (10), since the frequency of the injected voltage signal is much larger than the fundamental frequency, the fundamental component can be ignored and the $R_s$ can be ignored. Taking Equation (11) into (10), it can be gained that

$$\begin{cases} u_{\alpha h} = [L_0 + L_2\cos(2\theta)]\frac{di_{\alpha h}}{dt} + L_2\sin(2\theta)\frac{di_{\beta h}}{dt} \\ u_{\beta h} = L_2\sin(2\theta)\frac{di_{\alpha h}}{dt} + [L_0 - L_2\cos(2\theta)]\frac{di_{\beta h}}{dt} \end{cases} \tag{12}$$

### 2.3. Current Response under the High Frequency

When high-frequency voltage is injected into the $\alpha$-$\beta$-axis, the high-frequency current response can be obtained to estimate rotor position. Making $u_{\alpha h} = u_{\beta h} = U_m\cos\omega_h t$ and taking this into Equation (12), then high-frequency response current can be calculated as

$$\begin{bmatrix} i_{\alpha h} \\ i_{\beta h} \end{bmatrix} = \frac{U_m}{w_h(L_0^2 - L_2^2)} \begin{bmatrix} L_0 - \sqrt{2}L_2\cos(2\theta - \pi/4) \\ L_0 - \sqrt{2}L_2\sin(2\theta - \pi/4) \end{bmatrix} \times \sin w_h t \tag{13}$$

where $U_m$ is the amplitude of the high-frequency voltage and $\omega_h$ is the angular frequency of the high-frequency voltage. Equation (13) shows that $i_{\alpha h}$ and $i_{\beta h}$ are sinusoidal functions, that their amplitudes contain position angle information and that the amplitudes of $i_{\alpha h}$ and $i_{\beta h}$ are regulated by position angle.

In order to obtain the current amplitudes $I_\alpha$ and $I_\beta$, the traditional method is to multiply the values of $i_{\alpha h}$ and $i_{\beta h}$ by $\sin\omega_h t$ and then get the values of $I_\alpha$ and $I_\beta$ through a low-pass filter, as shown in Equation (14).

$$\begin{bmatrix} I_{\alpha h} \\ I_{\beta h} \end{bmatrix} = \begin{bmatrix} L_{PF}(i_{\alpha h}\sin w_h t) \\ L_{PF}(i_{\beta h}\sin w_h t) \end{bmatrix} = \frac{U_m}{2w_h(L_0^2 - L_2^2)} \begin{bmatrix} L_0 - \sqrt{2}L_2\cos(2\theta - \pi/4) \\ L_0 - \sqrt{2}L_2\sin(2\theta - \pi/4) \end{bmatrix} \tag{14}$$

where $L_{PF}$ refers to the low-pass filtering algorithm, and the results correspond to the current response amplitudes $I_\alpha$ and $I_\beta$ minus the direct-current(DC) component. The following Equation (15) is obtained (refer to Appendix A for the detailed derivation process).

$$\begin{bmatrix} I_{\alpha\theta} \\ I_{\beta\theta} \end{bmatrix} = k \begin{bmatrix} -\cos(2\theta - \pi/4) \\ -\sin(2\theta - \pi/4) \end{bmatrix} \tag{15}$$

where $k$ is positive, it can be expressed as

$$k = \frac{\sqrt{2}L_2 U_m}{2w_h(L_0^2 - L_2^2)} \tag{16}$$

### 3. Current Response to Calculate Rotor Position

*3.1. Traditional Method of Angle Extraction*

From the above section, after the high-frequency voltage signal is injected into the α-β-axis, the response current contains the initial rotor position information and $\theta$ can be extracted from it, as shown in Figure 3. The relation between the high-frequency response current and $\theta$ is obtained by Equation (15).

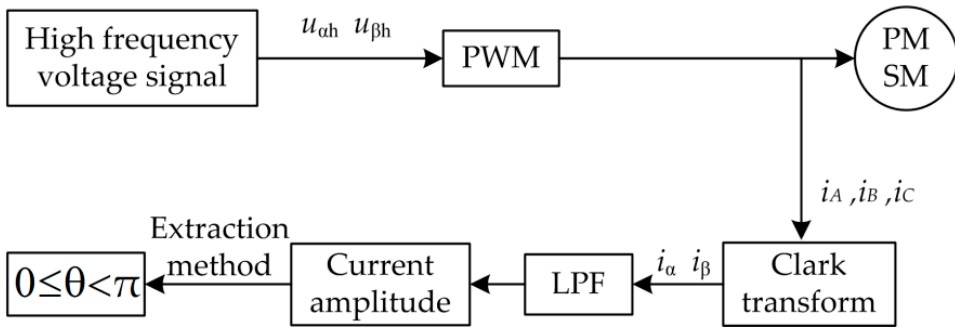

**Figure 3.** The extraction of rotor position.

In the Equation (14), $I_{\alpha\beta h}$ contains the position information of the rotor. It is necessary to extract the high-frequency current component $I_{\alpha\beta h}$ with a low-pass filter (LPF) or bandpass filter (BPF). However, the use of the LPF or BPF will affect the accuracy of the estimation. For example, a high-frequency signal of 500 Hz is extracted by BPF. The type of BPF is a typical second-order bandpass filter and the cut-off frequency is 500. The phase-frequency characteristic curve of the BPF, designed according to the above requirements, is shown in Figure 4.

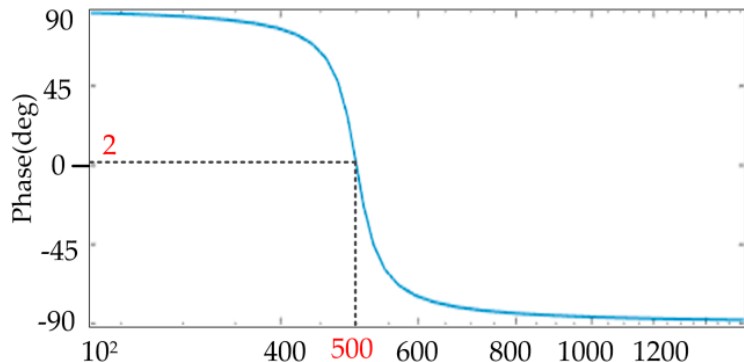

**Figure 4.** Phase-frequency characteristic curve of bandpass filter (BPF).

From the phase-frequency characteristic curve, it can be seen that a high-frequency signal passing through the BPF produces a phase shift of about 2° at 500 Hz. This phase shift also changes as the order of the filter, the cut-off frequency, the passband bandwidth and the frequency of the injected high-frequency signal change. Therefore, the use of a bandpass filter causes a phase shift in the signal and a bias in the estimated angle, which is one of the main problems to be solved below.

*3.2. Proposed Angle Extraction Method*

As can be seen from the above section, after injecting the high-frequency voltage signal into the α-β-axis we can get the response current, and the amplitudes of the response current, $I_\alpha$ and $I_\beta$, contain the information of the rotor position angle θ. The traditional method uses a low-pass filter to obtain the response current amplitudes $I_\alpha$ and $I_\beta$, while the low-pass filter will cause a delay in the phase angle and cause errors. These errors also need a corresponding compensation algorithm,

which increases the calculation amount of the system. Therefore, in order to reduce the errors caused by the use of the filter, the method of peak sampling is used; when $\sin\omega_h t = 1$, the values of $i_{\alpha h}$ and $i_{\beta h}$ are equal to $I_\alpha$ and $I_\beta$, and the current amplitudes of $I_\alpha$ and $I_\beta$ are obtained from Equation (17). This method does not need to use a low-pass filter and avoids the generation of phase delay.

$$\begin{bmatrix} I_\alpha \\ I_\beta \end{bmatrix} = \begin{bmatrix} i\alpha\left(wht = \frac{\pi}{2} + 2n\pi\right) \\ i_\beta\left(wht = \frac{\pi}{2} + 2n\pi\right) \end{bmatrix} = \frac{U_m}{2w_h\left(L_0^2 - L_2^2\right)} \begin{bmatrix} L_0 - \sqrt{2}L_2\cos(2\theta - \pi/4) \\ L_0 - \sqrt{2}L_2\sin(2\theta - \pi/4) \end{bmatrix} \tag{17}$$

After $I_\alpha$ and $I_\beta$ are obtained, the DC component is subtracted, and a component that is only related to the position angle $\theta$ can be obtained from Equation (15). The system block diagram is shown in Figure 5.

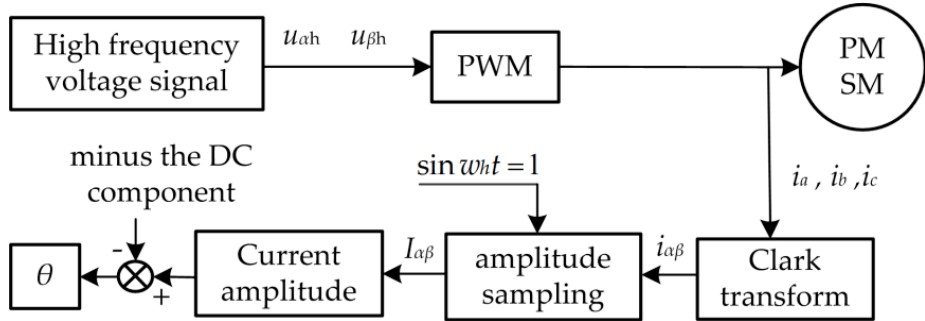

**Figure 5.** The extraction of rotor position.

According to Equation (15), the relationship between high-frequency response current and $\theta$ is shown in Figure 6.

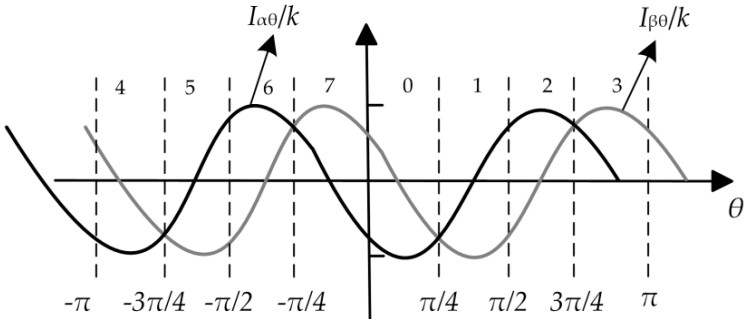

**Figure 6.** Relationship between the amplitude of high-frequency current and $\theta$.

From Equation (15) and Figure 6, $\theta$ can be estimated through the following formula:

$$\theta = \left[-\arcsin(I_{\beta\theta}/k) + \pi/4\right]/2 \tag{18}$$

According to the size of $I_{\alpha\theta}$ and $I_{\beta\theta}$, the interval of $\theta$ can be determined, and the formula can be used to estimate $\theta$ by a look-up table method. For example, $\theta$ can be estimated from the zero interval in Figure 5.

However, there are some inadequacies in the method of estimating $\theta$ according to the trigonometric function: since the $k$ contains $U_m$ and $\omega_h$, the position angle estimation will be affected by these parameters. If the motor parameters are not accurate, then the calculated initial angle will also have errors. By Equation (15), $I_{\beta\theta}/I_{\alpha\theta} = -\tan(2\theta - \pi/4)$, and there is $\theta = [\pi + \arctan(I_{\beta\theta}/I_{\alpha\theta}) + \pi/4]/2$. By using the arctan, the influence caused by the change of $k$ value will be reduced, and the calculation of $\theta$ will be more accurate. At the same time, when $I_{\beta\theta}/I_{\alpha\theta}$ is infinitely large in a special position, it needs to be linearized, solved by the following linearization method and without $k$.

Figure 7 is obtained by inversion of $I_{\alpha\theta}$ and $I_{\beta\theta}$ in Figure 6. From Figure 7, the relationship between the magnitude of the current response amplitude and the interval of $\theta$ is given by

$$\begin{cases} I_{\alpha-\theta} > I_{\beta\theta} > I_{\alpha\theta}, & 0 < \theta < \pi/4 \\ I_{\beta-\theta} > I_{\alpha\theta} > I_{\beta\theta}, & \pi/4 < \theta < \pi/2 \\ I_{\alpha\theta} > I_{\beta\theta} > I_{\alpha-\theta}, & \pi/2 < \theta < 3\pi/4 \\ I_{\beta\theta} > I_{\alpha\theta} > I_{\beta-\theta}, & 3\pi/4 < \theta < \pi \end{cases} \tag{19}$$

where $I_{\alpha-\theta}$ and $I_{\beta-\theta}$ are, respectively, the inverse of $I_{\alpha\theta}$ and $I_{\beta\theta}$, as shown in Figure 7. The case of $\pi<\theta<2\pi$ is same as that of $0<\theta<\pi$, and the next section introduces the determination of the two cases. First, the interval of $\theta$ is determined by Equation (19), and then $\theta$ is calculated according to the arctan value. When the value of $I_{\beta\theta}/I_{\alpha\theta}$ between 0 and $\pi$ is infinitely large, that is, $I_{\alpha\theta}$ is zero, then the corresponding $\theta$ has two values, $3\pi/8$ and $7\pi/8$. Using the linearized formula, we can calculate the angle around the two values within 1 electrical angle, and the other values can be calculated by the inverse tangent method.

$$\begin{cases} \theta = \left[\pi + \arctan(I_{\beta\theta}/I_{\alpha\theta}) + \frac{\pi}{4}\right]/2, & (I_{\alpha-\theta} > I_{\beta\theta} > I_{\alpha\theta}) \text{ or } (I_{\beta-\theta} > I_{\alpha\theta} > I_{\beta\theta}) \\ \theta = \left[\arctan(I_{\beta\theta}/I_{\alpha\theta}) + \frac{\pi}{4}\right]/2, & (I_{\alpha\theta} > I_{\beta\theta} > I_{\alpha-\theta}) \text{ or } (I_{\beta\theta} > I_{\alpha\theta} > I_{\beta-\theta}) \end{cases} \tag{20}$$

In the formula, the value of arctangent is taken between $(-\pi/2, \pi/2)$, and the value of $\theta$ in the formula is within an electric angle of $3\pi/8$ and $7\pi/8$, which is calculated according to the following linearization formula.

$$\begin{cases} \theta = \frac{I_{\beta-\theta}-I_{\alpha-\theta}}{I_{\beta-\theta}-I_{\beta\theta}} \times \frac{\pi}{4} + \frac{\pi}{4}, & \frac{3\pi}{8} - \frac{\pi}{180} < \theta < \frac{3\pi}{8} + \frac{\pi}{180} \\ \theta = \frac{I_{\beta\theta}-I_{\alpha\theta}}{I_{\beta\theta}-I_{\beta-\theta}} \times \frac{\pi}{4} + \frac{3\pi}{4}, & \frac{7\pi}{8} - \frac{\pi}{180} < \theta < \frac{7\pi}{8} + \frac{\pi}{180} \end{cases} \tag{21}$$

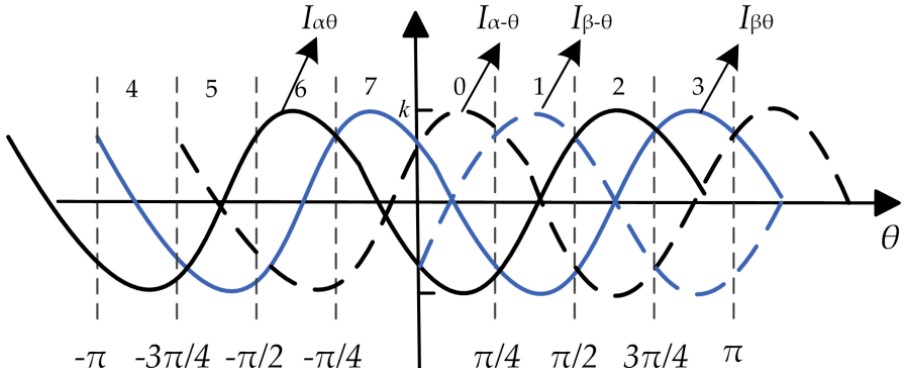

**Figure 7.** Amplitude of high-frequency current and its reverse.

The linearized calculation method can calculate the error angle, and this can be calculated as Equation (22).

$$\Delta\theta = \theta - \frac{I_{\beta-\theta} - I_{\alpha-\theta}}{I_{\beta-\theta} - I_{\beta\theta}} \times \frac{\pi}{4} - \frac{\pi}{4} \tag{22}$$

We take the linearization formula near $3\pi/8$ as an example, where $\theta$ is the actual angle of the motor, and Equations (22) and (15) are combined to get the following formula

$$\Delta\theta = \theta - \frac{K\pi}{4} \frac{-\cos(2\theta)}{\sin(2\theta) - \cos(2\theta)} - \frac{\pi}{4} \tag{23}$$

When $\theta = 3\pi/8$, it is taken into the above formula, then the calculation error is less than 0.68 electric angles. This shows that the position angle obtained by the linear calculation method has

minimal error and the effect of the error on the start-up of IPMSM can be ignored. At the same time, the error is produced only on small interval of $\theta$ in Equation (21), and has no effect on the other positions.

### 3.3. Angle Error Analysis

When the rotor position is at a special angle, the switching error analysis for the arctangent method and linear formula method is as follows.

According to Section 3.1 and Equations (20) and (21), when the initial position of the motor is near a special angle, for example, $3\pi/8$ and $7\pi/8$, the switching between Equations (20) and (21) is required. When arctan $(I_{\beta\theta}/I_{\alpha\theta})$ is less than 89° (electrical angle), the position angle is calculated by Equation (20). When arctan $(I_{\beta\theta}/I_{\alpha\theta})$ is greater than 89° (electrical angle) and less than 91° (electrical angle), the position angle is calculated by Equation (21). When the formula is switched, the position angle will produce an error. For example, when $\theta$ is $3\pi/8$, the switching errors of Equations (20) and (21) can be expressed by Equation (23).

$$\Delta\theta = \left[\arctan\left(I_{\beta\theta}/I_{\alpha\theta}\right) + \frac{\pi}{4}\right]/2 - \left(\frac{I_{\beta-\theta} - I_{\alpha-\theta}}{I_{\beta-\theta} - I_{\beta\theta}} \times \frac{\pi}{4} + \frac{\pi}{4}\right) \tag{24}$$

According to Equation (24), the angle error of formula switching is 0.1° (electrical angle), and the error of the above linear Equation (21) has been calculated before, so when Equations (20) and (21) are switched, the influence of angle calculation can be ignored. When $\theta$ is $3\pi/8$, the simulation diagram of angle error is shown in Figure 8. It can be seen from the figure that the error brought by formula switching is less than 0.1° (electrical angle), which is not much different from the theoretical calculation. This shows that the influence of angle error brought by formula switching can be ignored.

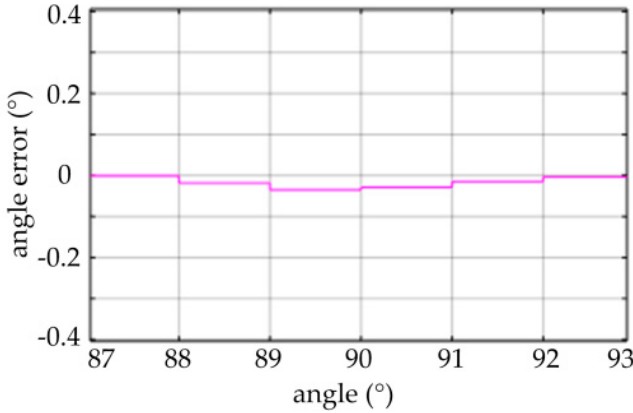

**Figure 8.** Angle error.

The influence of $U_m$ is analyzed as follows.

As mentioned before, in the traditional method, arcsin $(I_{\beta\theta}/k)$ is used to calculate $\theta$ in Equation (15). It can be seen that the Equation has coefficient $k$, and $k$ contains physical quantities such as $U_m$. The high-frequency voltage contains high-order harmonic components, which will bring errors to the calculation results. According to Equation (16), we have

$$\arcsin\left(I_{\beta\theta}/k\right) = \arcsin\left(\frac{\sqrt{2}w_h\left(L_0^2 - L_2^2\right)I_{\beta\theta}}{L_2 U_m}\right) \tag{25}$$

Based on the theoretical analysis of the influence of voltage $U_m$, taking the 20 V high-frequency voltage as an example, the angle is calculated by using the arcsin formula and the formula is $\theta = [-\arcsin (I_{\beta\theta}/k) + \pi/4]/2$. When $\theta = \pi/4$, in combination with Equation (15), arcsin(−0.707) can be calculated. Then combining Equation (24), arcsin(−0.707)=arcsin(−14.14/20). The voltage contains high-order harmonics. Taking the third harmonic as an example, the amplitude of the third harmonic

is $U_{m3}$ = 20/3 V = 6.67 V. The sum of $U_m$ and $U_{m3}$ is 26.67 V. Finally, the actual voltage injected into the $\alpha$-$\beta$-axis will become 26.67 V, so value of arcsin will become arcsin($-14.14/26.67$). After calculation, there is a 6.5° (electrical angle) deviation between $\theta$ and $\pi/4$, which has a great influence.

## 4. Polarity Judgment

It can be seen from the above section that when extracting the rotor position information from the high-frequency current response signal, due to the symmetry of the rotor salient pole, only 0 to $\pi$ can be estimated, and the actual positive direction of the d-axis cannot be determined; that is, the rotor polarity cannot be determined. Therefore, polarity judgment is needed. The principle of polarity judgment is that the stator current generates a magnetic field $\psi_s$, and the rotor permanent magnet generates $\psi_f$. When the two magnetic fields are in the same direction, this will have a magnetic aid effect on the synthetic magnetic field $\psi$ of the air gap, as shown in Figure 9a, increasing the magnetic saturation degree of the stator core, and entering the saturation area, as shown in Figure 9c. As the magnetic resistance increases, the winding inductance will decrease, and the winding current and inductance increase. In contrast, the winding current will increase; when $\psi_s$ and $\psi_f$ are in opposite directions, demagnetization will occur, as shown in Figure 9b, the magnetic saturation of the stator core will weaken and return to the linear area, as shown in Figure 9d, the magnetoresistance will decrease, the magnetoresistance inductance will increase, and the winding current will decrease.

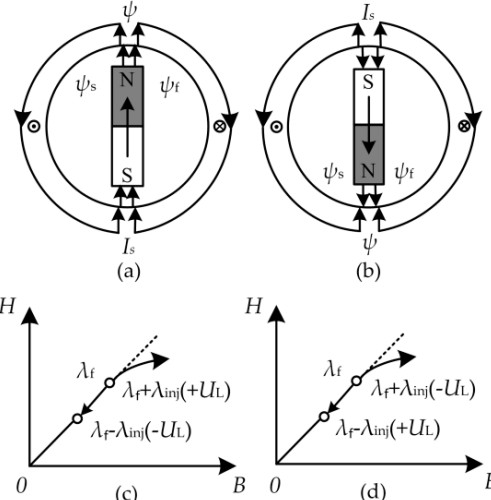

**Figure 9.** (**a**,**b**) represent the distribution of the air gap magnetic field; (**c**,**d**) represent the principle of rotor pole polarity.

The method steps of polarity judgment are as follows.

(1)　A position angle $\theta$ has been calculated in the previous section, and a constant voltage $U_L$ has been injected along the angle of $\theta$. The d-axis will generate a response current in this period of time, and reach the peak value. Remember the current value is $i_{d,\theta}$, and its absolute value is $|i_{d,\theta}|$.
(2)　Make $U_L$ = 0, and reduce the d-axis current to zero, so as to avoid affecting the following steps.
(3)　Inject the voltage $U_L$ of the same amplitude and time along the direction of $\theta+\pi$, record the peak value of d-axis response current as $i_{d,\theta+\pi}$ and record the absolute value as$|i_{d,\theta+\pi}|$.
(4)　Compare $|i_{d,\theta}|$ and $|i_{d,\theta+\pi}|$, if $|i_{d,\theta}| > |i_{d,\theta+\pi}|$, then prove that $\theta$ is the direction of the N-pole and $\theta$ is the actual angle, otherwise, prove that $\theta$ is the direction of the S-pole and $\theta + \pi$ is actual angle.

When the rotor position angles are $\theta$ and $\theta + \pi$, a voltage with amplitude $U_L$ is injected into the d-axis

$$\begin{bmatrix} \frac{di_d}{dt} \\ \frac{di_q}{dt} \end{bmatrix} = \begin{bmatrix} \frac{1}{L_d} & 0 \\ 0 & \frac{1}{L_q} \end{bmatrix} \begin{bmatrix} u_L(t) \\ 0 \end{bmatrix} - R_s \begin{bmatrix} i_d \\ i_q \end{bmatrix} \tag{26}$$

where $u_\text{L}(t)$ is the step variable with time and its maximum value is $U_\text{L}$. Since $R_\text{S}$ is very small, it is not included, and Equation (26) can be reduced to

$$\frac{di_\text{d}}{dt} = \frac{u_\text{L}(t)}{L_d} \tag{27}$$

Equation (26) shows that when the step process is over, $u_\text{L}(t)$ is equal to $U_\text{L}$, and there is an inverse relationship between the maximum values of $i_\text{d}$ and $L_\text{d}$. In this way, the amplitude of the current response is obtained by injecting the high-frequency voltage into the d-axis, then recording the d-axis current amplitudes, $|i_{d,\theta}|$ and $|i_{d,\theta+\pi}|$, when the position angles are $\theta$ and $\theta+\pi$ respectively. It can be gained that

$$\begin{cases} \theta, & |i_{d,\theta}| > |i_{d,\theta+\pi}| \\ \theta + \pi, & |i_{d,\theta}| < |i_{d,\theta+\pi}| \end{cases} \tag{28}$$

The system diagram of polarity judgment is shown in Figure 10.

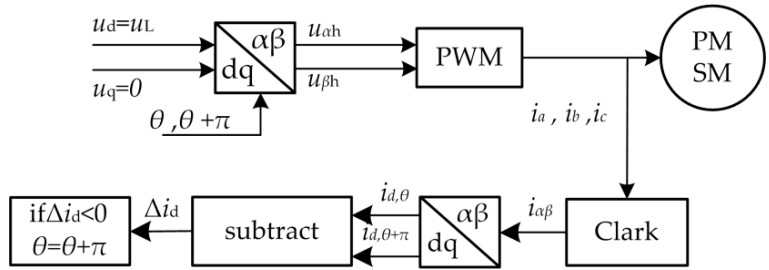

**Figure 10.** Relationship of stator magnetic potential and flux.

## 5. Experimental Results

In this study, the theoretical correctness of the rotor initial position detection was verified by experiment. An experimental platform for detecting initial rotor position of permanent magnet synchronous motor was set up, as shown in Figure 11. In the experimental system, an interior permanent magnet synchronous motor with a rated power of 20 kW was selected as the research object and the rotor had a salient pole structure. The parameters of the motor are shown in Table 1. The core of the control system was a TMS320F28335, the main circuit was a voltage type three-phase inverter and the switching frequency was 10 kHz. The amplitude of high-frequency voltage injected into permanent magnet synchronous motor was 20 V, and the frequency was 500 Hz.

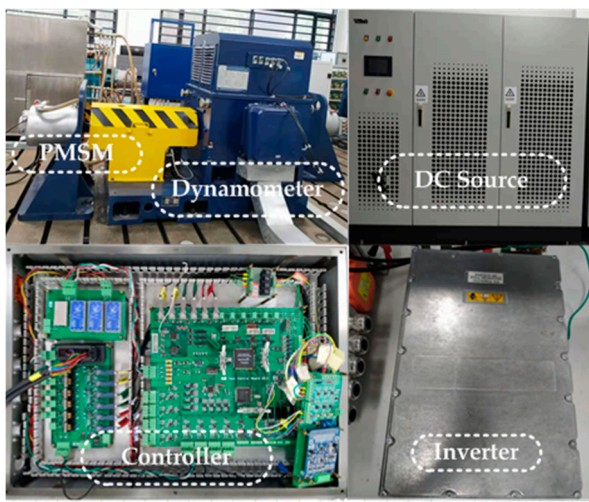

**Figure 11.** The experimental system.

**Table 1.** The simulation parameters of the PMSM.

| $U_N$/V | $P_N$/kW | $T_N$/N·m | $n$/r/min | $J$/kg·m$^2$ | $L_d$/mH | $L_q$/mH | $P$ | $R_s$/Ω | $\psi_f$/Wb |
|---|---|---|---|---|---|---|---|---|---|
| 300 | 20 | 64 | 3000 | 0.0033 | 0.2 | 0.5 | 4 | 0.01023 | 0.071 |

Figure 12a shows the actual and estimated positions of the rotor when the actual rotor angle is 88.7°. In the figure, $\theta_{rel}$ is the true rotor position obtained by the resolver, $\theta_{cal}$ is the angle obtained by the improved algorithm, $i_d$ is the current response when the positive direction injection and the reverse direction injection voltage are constant when the polarity is judged, and $\Delta i_d$ is the difference in the current response.

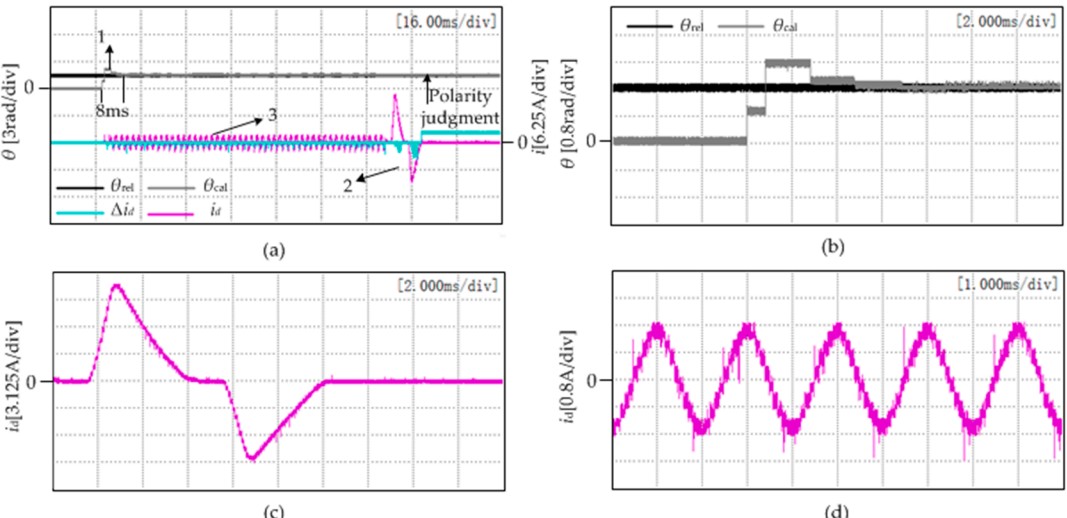

**Figure 12.** When $\theta$ = 88.7°: (**a**) angle comparison and d-axis current experimental waveforms; (**b**) amplified waveform of 1 area from Figure 12a; (**c**) amplified waveform of 2 area from Figure 12a; (**d**) amplified waveform of 3 area from Figure 12a.

In Figure 12b, the calculation of the position angle takes about 8 ms. In Figure 12c, When the polarity is judged, the positive value of $\Delta i_d$ indicates that the current response during the d-axis forward voltage injection is larger than that in the negative injection, indicating $|i_{d,\theta}|>|i_{d,\theta+\pi}|$; therefore, it can be judged that the rotor position is at the N-pole, $\theta = \theta_{cal}$. Figure 12d shows the current waveform of $i_d$ when injecting the high-frequency signal.

As shown in Figure 13a, the actual position of the rotor was 307.33°. The rotor position is calculated after the polarity is determined. As shown in Figure 13b, the calculation of the position angle takes about 8 ms.

As can be seen from Figure 13a, the actual position of the rotor was 129.485° and the rotor position calculated before the polarity judgment is different from the actual position angle by $\pi$. In Figure 13a, when the polarity is judged, $\Delta i_d$ is negative, indicating that the current response when the d-axis forward voltage is injected is smaller than that of the negative injection, which is $|i_{d,\theta}|<|i_{d,\theta+\pi}|$; therefore, it can be judged that the rotor position is at the S-pole, $\theta = \theta_{cal} + \pi$. Figure 13d shows the current waveform of $i_d$ when injecting a high-frequency signal.

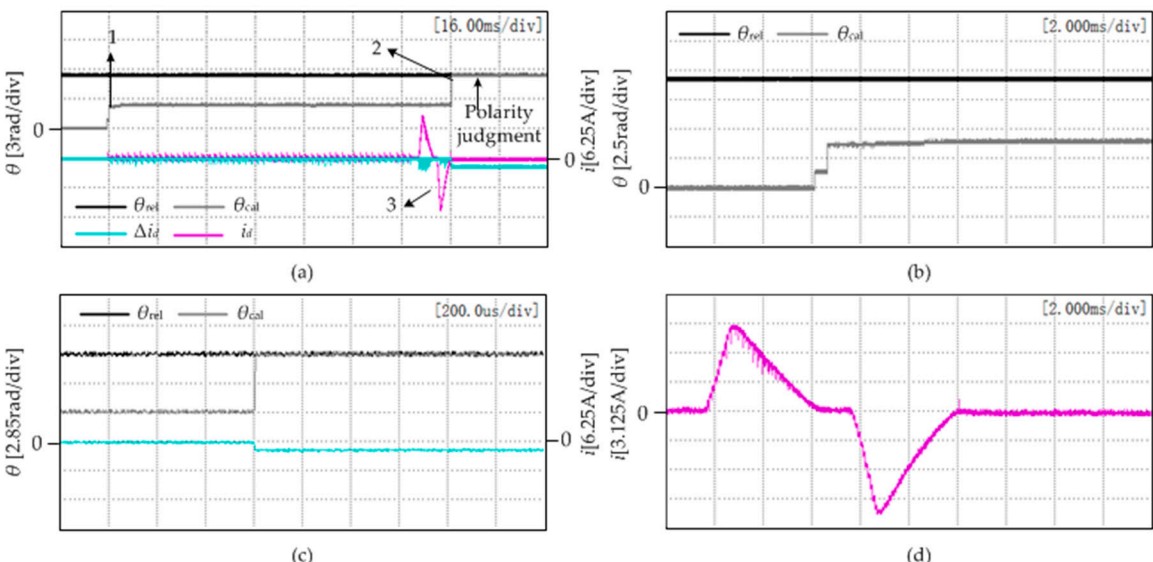

**Figure 13.** When θ = 307.33°: (**a**) angle comparison and d-axis current experimental waveforms; (**b**) amplified waveform of 1 area from Figure 12a; (**c**) amplified waveform of 2 area from Figure 12a; (**d**) amplified waveform of 3 area from Figure 12a.

Figure 14a shows the high-frequency current response waveform when $\theta = 88.7°$. Figure 14a shows the $i_\alpha$, $i_\beta$ instantaneous values and the amplitudes $I_\alpha$, $I_\beta$ containing the direct current component. When the rotor position is less than 90°, $i_\alpha$ is asymmetrical due to the mechanical asymmetry of the motor. In Figure 14, the current amplitude obtained by the improved method is very accurate. The direct current component is calculated to be 25.25A, and after the direct current component is subtracted, $I_{\alpha\theta} = -9.63$A, $I_{\beta\theta} = 9.135$A and the interval in which the current is located, that is, the interval $I_{\alpha-\theta} > I_{\beta\theta} > I_{\alpha\theta}$, and $\theta = 90.765°$ can be obtained according to Equation (19). The error is mainly caused by the inaccuracy of current sampling. During the operation of the motor, the inductance will reverse, which leads to the difference between the current value and the theoretical value. This includes the fact that the DC component fluctuations ripple at a relatively accurate value in each experiment, which leads to errors in the values of $I_{\alpha\theta}$ and $I_{\beta\theta}$ because the motor operates at zero speed and no load, so this the error will be very small. Comparing the result estimated by the formula with the actual result, it can be concluded that the error of position estimation is 2.07°.

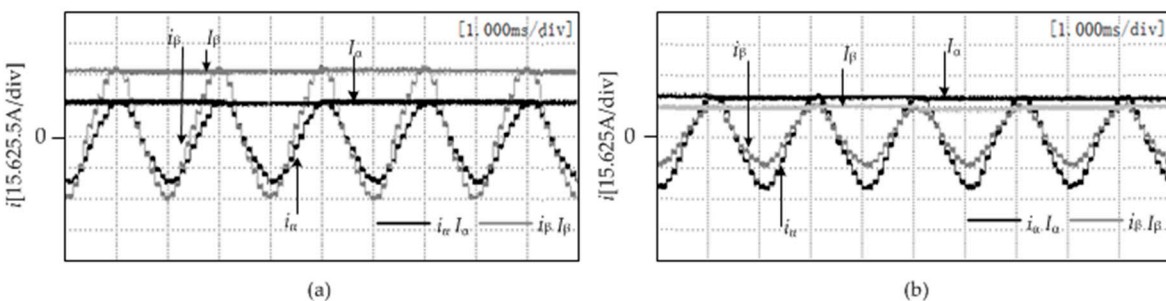

**Figure 14.** High-frequency response current: (**a**) θ = 88.7°; (**b**) θ = 307.33°.

Figure 14b shows the high-frequency current response waveform when $\theta = 307.33°$. After subtracting the direct current component, $I_{\alpha\theta} = -9.625$A, $I_{\beta\theta} = -6.49$A, it can be judged that the current is in the interval, where $I_{\alpha-\theta} > I_{\beta\theta} > I_{\alpha\theta}$, and $\theta = 129.485°$ can be obtained according to Equation (19). Since $\theta$ can only calculate the value within $\pi$, as shown in Figure 13d, the polarity can be judged by polarity judgment. The position angle should be added to $\pi$, and the estimated position angle is 309.485°. Compared with the actual position, the position estimation error is 2.15°.

Figure 15 shows the comparison between the estimated value and the actual value of the rotor position. By changing the actual initial position of the motor, the actual initial position of the motor is increased from 0 rad to $2\pi$ rad at about every 0.55 rad interval. Compared with the calculated initial position of the motor, Figure 15 is obtained. It can be seen from the figure that the maximum error is within $5°$, the average error is approximately $2.7°$ and the detection error of the rotor position is relatively small. It has little effect on the start-up of the PMSM.

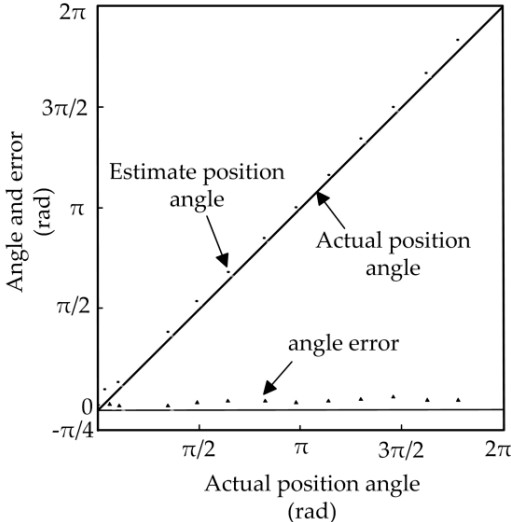

**Figure 15.** Comparison between actual angle and estimated angle.

Although the accuracy of the initial position angle calculated by the traditional filter method is not different from that obtained by the proposed method, when the motor rotates, the method using the filter has a significant angle delay. This paper does not use the filter, so there is no angle delay. The experimental results are shown in Figure 16.

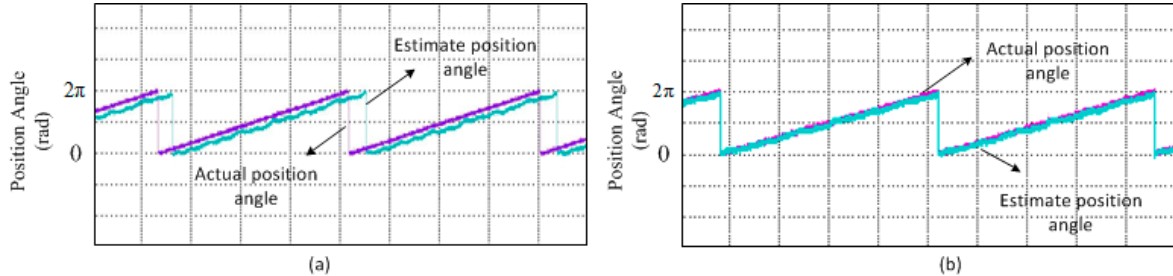

**Figure 16.** Angle comparison waveform after rotation: (**a**) traditional method (using filters); (**b**) method proposed in this paper.

From the above experimental results, it can be seen that the difference between the estimation of the rotor position detection value and the actual value is also small. The causes of the error are mainly the current sampling error, asymmetry of the motor structure, external disturbance of the system structure and current response error caused by different saturation degrees of the rotor in different positions. The current response has been averaged over several cycles to reduce the angular error of the rotor. In subsequent research, the number of periods of calculation can be increased to reduce angle error.

## 6. Conclusions

A rotor position detection method for a permanent magnet synchronous motor (PMSM) based on high-frequency voltage signal injection has been proposed in this paper. The rotating high-frequency

voltage signal is injected into the stator winding. According to the relation between the amplitude of high-frequency current response and rotor position angle, the rotor position information is obtained using inverse tangent and linear formulae, then the polarity of the rotor is distinguished from the variation of the inductance.

This method is simple and easy to implement, does not depend on parameters and does not need to use a low-pass filter. It has good application value and can be used in engineering practices. This paper is mainly applied to the IPMSM. The main mathematical model is also for the IPMSM and has not been extended to the surface permanent magnet synchronous motor (SPMSM), which is a limitation of this paper. Later, the model will be analyzed and verified for the surface mounted motor. In addition, the high-frequency sinusoidal signal injection method adopted in this paper, which is limited in frequency, will affect the dynamic response performance of the system, and increase the interference between the fundamental and high-frequency signals.

**Author Contributions:** Conceptualization, Z.W., X.L. and H.W.; methodology, Z.W. and B.Y.; software, B.Y. and X.J.; validation, H.W. and X.L.; formal analysis, Z.W. and B.Y.; writing—original draft preparation, B.Y.; writing—review and editing, Z.W. and L.G. and funding acquisition, Z.W. All authors have read and agreed to the published version of the manuscript.

**Funding:** This research was funded by "The Fund Projects of National Natural Science Foundation of China, grant number 51977150", "The Young Fund Projects of National Natural Science Foundation of China, grant number 51907142" and "The Young Fund Projects of Tianjin Natural Science Foundation of China, grant number 18JCQNJC74400", "The Natural Science Foundation of Tianjin (Grant No.18JCQNJC74200)", "The Science & Technology Development Fund of Tianjin Education Commission for Higher Education. (Grant No. 2018KJ208)".

**Conflicts of Interest:** The authors declare no conflict of interest.

## Appendix A

It can be seen from Equation (13) that the current is a sine function of $\omega_h$, and the following changes for Equation (13).

$$\begin{bmatrix} i_{\alpha h} \\ i_{\beta h} \end{bmatrix} \times \sin w_h t = \frac{U_m}{w_h\left(L_0^2 - L_2^2\right)} \begin{bmatrix} L_0 - \sqrt{2}L_2\cos(2\theta - \pi/4) \\ L_0 - \sqrt{2}L_2\sin(2\theta - \pi/4) \end{bmatrix} \times \sin^2 w_h t \tag{A1}$$

Make a trigonometric change to Equation (A1) to get the following formula.

$$\begin{bmatrix} i_{\alpha h} \\ i_{\beta h} \end{bmatrix} \times \sin w_h t = \frac{U_m}{w_h\left(L_0^2 - L_2^2\right)} \begin{bmatrix} L_0 - \sqrt{2}L_2\cos(2\theta - \pi/4) \\ L_0 - \sqrt{2}L_2\sin(2\theta - \pi/4) \end{bmatrix} \times \frac{1 - \cos 2w_h t}{2} \tag{A2}$$

Expand Equation (A2) into two items, which are

$$\begin{bmatrix} i_{\alpha h} \\ i_{\beta h} \end{bmatrix} \times \sin w_h t \quad = \frac{U_m}{w_h\left(L_0^2 - L_2^2\right)} \begin{bmatrix} L_0 - \sqrt{2}L_2\cos(2\theta - \pi/4) \\ L_0 - \sqrt{2}L_2\sin(2\theta - \pi/4) \end{bmatrix} \times \frac{1}{2}$$
$$- \frac{U_m}{w_h\left(L_0^2 - L_2^2\right)} \begin{bmatrix} L_0 - \sqrt{2}L_2\cos(2\theta - \pi/4) \\ L_0 - \sqrt{2}L_2\sin(2\theta - \pi/4) \end{bmatrix} \times \frac{\cos 2w_h t}{2} \tag{A3}$$

It can be seen from Equation (A3) that the DC component and high-frequency component are included. The DC component is not only the $I_\alpha$ and $I_\beta$, but the high-frequency component can also be filtered out through a low-pass filter to obtain DC components $I_\alpha$ and $I_\beta$, as follows

$$\begin{bmatrix} I_{\alpha h} \\ I_{\beta h} \end{bmatrix} = \begin{bmatrix} L_{PF}(i_{\alpha h}\sin w_h t) \\ L_{PF}(i_{\beta h}\sin w_h t) \end{bmatrix} = \frac{U_m}{2w_h\left(L_0^2 - L_2^2\right)} \begin{bmatrix} L_0 - \sqrt{2}L_2\cos(2\theta - \pi/4) \\ L_0 - \sqrt{2}L_2\sin(2\theta - \pi/4) \end{bmatrix} \tag{A4}$$

Expand the bracket in Equation (A4)

$$\begin{bmatrix} I_{\alpha h} \\ I_{\beta h} \end{bmatrix} = \begin{bmatrix} \mathrm{L_{PF}}(i_{\alpha h}\sin w_h t) \\ \mathrm{L_{PF}}(i_{\beta h}\sin w_h t) \end{bmatrix} = \frac{L_0 U_m}{2w_h(L_0^2 - L_2^2)} - \frac{U_m}{2w_h(L_0^2 - L_2^2)} \begin{bmatrix} \sqrt{2}L_2\cos(2\theta - \pi/4) \\ \sqrt{2}L_2\sin(2\theta - \pi/4) \end{bmatrix} \quad \text{(A5)}$$

It can be seen from (A5) that the right-hand side of the equals sign contains the DC component and the rotor position information. It is necessary to subtract the DC component to obtain the component related only to the rotor position information.

$$\begin{bmatrix} I_{\alpha\theta} \\ I_{\beta\theta} \end{bmatrix} = \frac{\sqrt{2}L_2 U_m}{2w_h(L_0^2 - L_2^2)} \begin{bmatrix} -\cos(2\theta - \pi/4) \\ -\sin(2\theta - \pi/4) \end{bmatrix} \quad \text{(A6)}$$

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
