# Peer review of "Initial Rotor Position Detection for Permanent Magnet Synchronous Motor Based on High-Frequency Voltage Injection without Filter"

_wevj, doi:10.3390/wevj11040071_

Round 1

Reviewer 1 Report

Comments and suggestions included directly in the paper

Author Response

Thank you very much for your detailed suggestions. According to the suggestions in your file, I have made some modifications and highlighted in yellow. The revised paper has been uploaded. Please check it.

Reviewer 2 Report

Comments for authors. 1. The authors should provide the comparison of the proposed technique with the conventional one. 2. Literature review is limited to conference papers. It must be extended to recent and relevant papers available in literature. 3. All figure are biller. It seems that authors have copied these figures from somewhere else, and pasted in this paper. 4. The proposed technique is basic. More relevant information about the implementation of the proposed technique must be provided. 6. Novelty is weak. Please emphasis on the novelty of the paper in the introduction and abstract of the paper. 7. Outcomes of the research may provided in the conclusion part.

Author Response

Thank you very much for your suggestion. We have modified the manuscript according to your advice and highlighted in yellow.

1. The authors should provide the comparison of the proposed technique with the conventional one.

Response 1:

Thank you very much for your advice. We have adjusted the structure of the paper, highlighting the difference between the traditional method and the method proposed in this paper.

2. Literature review is limited to conference papers. It must be extended to recent and relevant papers available in literature.

Response 2:

Thank you very much for your suggestion. We have adjusted the references and replaced most of the conference papers with journal papers published in recent years.

3. All figure are biller. It seems that authors have copied these figures from somewhere else, and pasted in this paper.

Response 3:

We can be sure that the pictures in the paper were created by ourselves, but we don't know why the quality of the pictures has deteriorated after uploading. We will pay attention to this problem this time.

4. The proposed technique is basic. More relevant information about the implementation of the proposed technique must be provided. 

Response 4:

Thank you very much for your suggestion. The innovation of this paper is not well reflected in the previous versions of the paper. This revision highlights the innovation of this paper.

5. Novelty is weak. Please emphasis on the novelty of the paper in the introduction and abstract of the paper.

Response 5:

The abstract is revised to further highlight the innovation of this paper, and the introduction and conclusion are also modified accordingly.

6. Outcomes of the research may provided in the conclusion part.

Response 6:

We modify the conclusion part and analyze the research results in the conclusion part.

Reviewer 3 Report

please improve the quality of the figures, the figures have the low qualities 

Could you please describe what is different between the technics provide in blow paper and your method?

Hyunbae Kim, Kum-Kang Huh, Robert D. Lorenz, and Thomas M. Jahns, ''A Novel Method for Initial Rotor Position Estimation for IPM Synchronous Machine Drives'', IEEE TRANSACTIONS ON INDUSTRY APPLICATIONS, VOL. 40, NO. 5, SEPTEMBER/OCTOBER 2004

Author Response

Thank you very much for your suggestion. We have modified the paper according to your advice and highlighted in yellow.

1. Please improve the quality of the figures, the figures have the low qualities.

Response 1:

I'm very sorry. Before uploading the paper, the figures in the paper are very clear, but the figures become blurred after uploading. We will pay attention to this problem this time. We have uploaded the word of the revised paper. And we think the revised paper would be clear enough. Thank you for your reminding.

2. Could you please describe what is different between the technics provide in blow paper and your method?

Hyunbae Kim, Kum-Kang Huh, Robert D. Lorenz, and Thomas M. Jahns, ''A Novel Method for Initial Rotor Position Estimation for IPM Synchronous Machine Drives'', IEEE TRANSACTIONS ON INDUSTRY APPLICATIONS, VOL. 40, NO. 5, SEPTEMBER/OCTOBER 2004.

Response 2:

In the above paper, Initial rotor position is estimated from a spatial saliency tracking estimator.  Magnet polarity is identified simultaneously by using the fundamental harmonic of rotor position in the second harmonic frame of the carrier frequency due to d and q-axis saturation saliencies for both rotating and pulsating voltage carrier injection. In his method, low-pass filter is inevitably used, which leads to a certain phase delay. In this paper, the proposed method avoids the use of filters, angle calculation is more accurate, and the amount of calculation is less.

Round 2

Reviewer 1 Report

If Ld and Lq are inductances, than Eqs. 25 and 26 are not thrue.

Please explain

Author Response

I am very sorry that the formulation of the formula is not accurate enough and it is a clerical error. We are sorry for the inconvenience. In the revision, we have modified it and relevant explanation has been added. Both of them has been highlighted in yellow in the revision.

Reviewer 2 Report

Authors have improved the revised paper considering the comments of the reviewer. The paper can now be published in its current form.

Author Response

Thank you very much for your review again.

Reviewer 3 Report

You mention ''In this paper, the proposed method avoids the use of filters, angle calculation is more accurate, and the amount of calculation is less.'' how you evaluate that your method is more accurate?

how much you improve the accuracy,

I have not seen any comparison between that method and your method. 

Author Response

I'm very sorry, because I didn't explain clearly in my last reply, which brought you wrong understanding. Compared with the traditional method, the proposed method mainly does not use a filter, and reduces the position angle estimation error by adding compensation link. According to your suggestion, we have supplemented the experiment with filter without compensation as soon as possible and compared with it.

As can be seen from the experimental results, the traditional method uses a filter, which will produce a certain angle delay, which is more obvious after the motor rotates; however, the method proposed in this paper does not use a filter, so there is no angle delay. The new experimental waveforms are also included in the revised paper and highlighted in yellow. Thank you for your comments.

Round 3

Reviewer 1 Report

I have no additional comments

Reviewer 3 Report

Thank you for your answer. .